# A Study on the Measurement Characteristics of the Spring-Plate Flow Measurement Device

**Xiaoniu Li, Siyuan Tao, Yongye Li * and Li Wan**

College of Water Resource Science and Engineering, Taiyuan University of Technology, Taiyuan 030024, China; 13734008771@163.com (X.L.); taosiyuan0603@link.tyut.edu.cn (S.T.); wanli_valley@163.com (L.W.)
* Correspondence: liyongye@tyut.edu.cn; Tel.: +86-139-3423-9832

**Abstract:** In order to improve the effective utilization of agricultural irrigation water and to reasonably allocate water resources in irrigation areas, it is necessary to use open channel flow measurement devices that are accurate and easy to carry. In this study, a spring-plate flow measurement device with different plate widths was designed. Through a combination of theoretical analysis and numerical simulation, the measurement characteristics of the device in specific channels under conditions of 20–105 m³/h flow were studied, the relationship between the flow rate and the force acting on the plate surface was fitted, and the hydraulic characteristics of water flow during its use, such as pressure, velocity distribution, and head loss, were analyzed. The results show that in the process of using the spring-plate flow measurement device, the force on the plate surface increases with the increase in the flow rate, and the force on the plate surface is related to the flow rate in the channel by a power of 5/6. The width of the measurement plate impacts the accuracy of flow measurement, and the smaller the plate width, the larger the error in flow measurement. The distribution of pressure on the measurement plate is similar to that of static pressure, and the pressure increases with the increase in the width of the measurement plate. The upstream flow velocity of the device is small, and the water level increases due to obstruction of the measurement plate. When it connects to the downstream water surface, the water level rapidly decreases, and the flow velocity increases. In using the spring-plate flow measurement device to measure flow, head loss will be produced, and the magnitude of this loss increases with the increase in the width of the measurement plate. The research results provide a theoretical basis for the application of spring plate flow-measuring devices in irrigation areas.

**Keywords:** spring plate; rectangular channel; channel flow measurement; hydraulic characteristics

## 1. Introduction

China is suffering from a serious shortage of water resources, and with the development of society and economy, China's water consumption has gradually increased, of which agricultural water has reached 61% of the total water consumption, thus representing the main way in which water is used [1,2]. However, the effective utilization of agricultural irrigation water in China's farmland is low, which indicates the waste of water resources and also affects the yield of crops [3,4], and water use should be planned on a scientific basis to ensure the rational management of irrigation water. Accurate measurement of water quantity in irrigation areas is an important measure to achieve rational allocation of water resources in irrigation areas, and it is of practical importance to design flow measurement devices for water distribution channels that produce accurate measurements and are easy to carry [5,6]. Traditional channel flow measurement methods have the problem of low accuracy [7,8], and the current flow measurement methods that have mainly been studied are the flow-measuring flume method and the novel flow-measuring equipment method.

The principle of the flow-measuring flume method is to install specific water measurement facilities in the water, according to the Venturi principle, and the water flow through

the contracted channel cross-section will produce critical flow, and the flow rate can be calculated by substituting the measured water depth and other results into the empirical formula. The design of the flow-measuring flume will directly affect the accuracy of the flow results. In a study of the flow-measuring flume, Samani et al. [9,10] constructed a simple form of a flow-measuring flume by placing semicircular cylinders on the side walls of a rectangular channel to constrict water flow, and the flow calculated by the formula was compared with the flow measured in field experiments, with the error determined to be less than 5%. Zhang et al. [11] studied the flow measurement process of a semi-cylindrical flow-measuring flume in rectangular channels under three different shrinkage ratios, and by using dimensional analysis, obtained a concise and practical flow equation with less error. Liu et al. [12,13] studied a column flow-measuring flume with a wing profile shape, which exerts less influence on the water flow pattern when the water flows through due to the streamlined boundary, and they analyzed the hydraulic characteristics of the wing column type flow-measuring flume and studied hydraulic parameters such as the water surface line and Fourdrinier number under different operating conditions. In addition, many scholars have also designed flow-measuring flumes with various forms, such as the cut-throat flow-measuring flume [14,15], triangular long-throat flow-measuring flume [16], and central baffle flow-measuring flume, and worked out the corresponding empirical formulas, which can be used for different types of flow measurement channels. The flow-measuring flume method has the advantages of simple device structure, low cost, and convenient flow calculation, but it is not suitable for channels with large sand content and will raise the water level of the channel and generate large head loss when used.

The novel flow-measuring equipment method is based on various physical principles, and the flow is measured using novel water measuring facilities such as ultrasonic flow-measuring equipment and pressure type flow-measuring equipment. Tamari et al. [17] designed a noncontact handheld radar flow meter that can quickly estimate the flow rate and be used to measure the flow state in difficult-to-reach streams. Wang et al. [18] designed a pressure-based open channel flow measurement device based on Bernoulli's principle, in which the pressure difference between two points can be measured by a pressure sensor, and the flow rate can then be deduced. Peng et al. [19] designed an automatic flow measurement device based on the flow velocity area method, in which the water level and flow velocity data are measured at each point of the channel cross-section by water level sensors and flow velocity sensors, so as to calculate the flow magnitude, and the flow measurement results of this automatic flow measurement device were found to be highly accurate. There is a rich variety of novel flow-measuring equipment methods, such as radar technology [20], STIV [21], the flow-velocity sensor [22], and ultrasonic transit time [23], which can be used for different types of channels, and the measurement accuracy is high. However, the structure of equipment used in this method is complex, and the cost is high.

It can be seen that for the selection of a channel flow measurement device, the requirements of applicability, economy, and flow measurement accuracy should all be simultaneously considered. Therefore, we have designed a spring plate flow measurement device to meet the advantages of a simple structure and low operating head loss. On the basis of laboratory model experiments, this study utilized the hydraulic characteristics of the flow-measuring device for numerical simulation, and compared and analyzed it with the results of model experiments. Under the working conditions of a 20–105 m$^3$/h flow rate, the relationship between the flow rate and the force on the plate surface is fitted. The fitted flow measurement formula is then analyzed for errors, and the hydraulic characteristics of the flow during measurement are studied.

## 2. Research Program and Model Construction

### 2.1. Device Structure and Flow Measurement Principle

The structure of the spring-plate flow measurement device is mainly composed of two parts: measurement core parts and additional fixed parts. Among them, the measurement

core components mainly include the measurement housing, measurement guide, compression spring, measurement sleeve, measurement plate, measurement slider, and slide rail. The additional fixed parts include a support bar and housing reinforcement. The structure of the spring-plate flow measurement device [24] is shown in Figure 1.

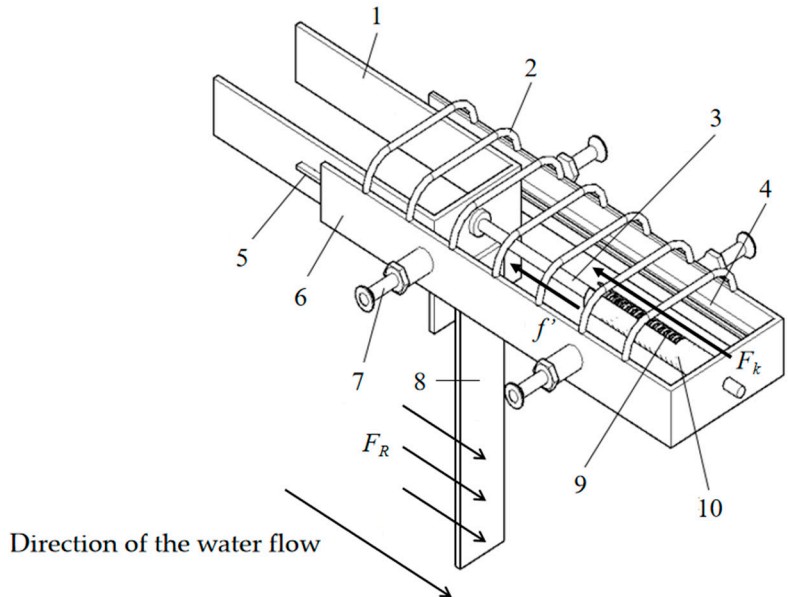

**Figure 1.** Schematic diagram of the structure of the spring-plate flow measurement device. 1: Measurement housing; 2: housing reinforcement; 3: measurement guide; 4: slide rail; 5: measurement slider; 6: housing; 7: support bar; 8: measurement plate; 9: compression spring; 10: measurement sleeve.

The measurement process is as follows. First, rotate and open the support rods located on both sides of the device to suspend it in the channel. Adjust the height so that the measuring plate is higher than the bottom of the channel, rotate and open the support rod at the other end, and at this time, the device is tilted and fixed in the channel; afterwards, repeat the above steps to fine tune the height of the device, and finally place the measuring plate at the bottom of the channel without affecting its longitudinal movement. Gently pull the measuring shell upstream so that the thin circular plate at the front end of the measuring guide rod lightly touches the spring. The water flow entering the channel impacts the measuring plate, causing the measuring plate to drive the measuring shell downstream. Therefore, the measuring guide rod compresses the spring placed in the spring cylinder and produces deformation. According to Hooke's law ($F_K = kd$), the spring force is obtained, thereby obtaining the magnitude of the force exerted on the measuring plate by the water flow.

The spring-plate flow measurement device mainly uses the interaction between the water flow and the measurement plate to measure the flow. In the process of measurement, the measurement plate of the spring-plate flow measurement device reduces the cross-sectional area of the channel, and the water flow is obstructed when passing through the measurement plate, resulting in a higher water level upstream of the measurement plate, which converts kinetic energy into potential energy and produces differential pressure resistance on both sides of the measurement plate. The pressure difference between the two sides will push the measurement plate along the slide direction of horizontal movement, thus squeezing the spring until the force of the water flow pushing the measurement plate is in balance with the spring force and the friction between the devices, that is:

$$F_R = F_k + f' \tag{1}$$

In the formula, $F_R$ is the force of the water flow pushing the measurement plate, $F_k$ is the spring force, and $f'$ is the friction between the devices.

The magnitude of the drag force $F_R$ on the measuring plate can be represented by Equation (2):

$$F_R = C_D \frac{\rho_w u_w^2}{2} A_D \tag{2}$$

In the formula, $C_D$ is the drag coefficient of the flow around, which can be determined by experiments and is mainly related to Reynolds number $Re$; $\rho_W$ is the density; $U_w$ is the relative velocity of the fluid and object in the channel before being affected by the surrounding flow; $A_D$ is the projected area of the surrounding object in the vertical direction of the flow.

The flow velocity $u_w$ in the channel is related to the average velocity $u$ of the channel section, indicating a direct relationship between $F_R$ and the average velocity $u$ of the channel section. Because the average velocity $u$ can be expressed as $u = Q/A$, there is a direct relationship between $F_R$ and $Q$.

The distance the spring is compressed can be read from the scale on the measurement sleeve, and the spring force $F_k$ can then be calculated according to Hooke's law. The frictional force $f'$ between the devices is a constant and can be obtained by experimental measurements. By adding the spring force and the frictional force between the devices together, the force $F_R$ of the water pushing the measurement plate can be obtained. Finally, the flow rate $Q$ in the channel can be calculated from the force $F_R$.

### 2.2. Geometric Modelling

The geometric model of the spring-plate flow measurement device was constructed using AutoCAD. The width of the rectangular channel was taken as 270 mm, and the bottom slopes were taken as 1/500, 1/1000, 1/2500, 1/5000, and 1/10,000. In order to study the effect of different plate widths on the measurement characteristics of the spring-plate flow measurement device, four different plate widths of 30, 40, 50, and 60 mm were modeled. Since only the measurement plate is in contact with the water body during flow measurement, the entire device can be simplified to a thin rectangular plate in a fixed position when constructing the model. In order to better simulate the flow pattern in the rectangular channel and smoothen water flow in the upstream and downstream of the measurement plate, the length of the channel model is set as 7 m, and the measurement plate is located at 2 m from the entrance of the channel. The geometric model of the channel and the flow measurement device is shown in Figure 2.

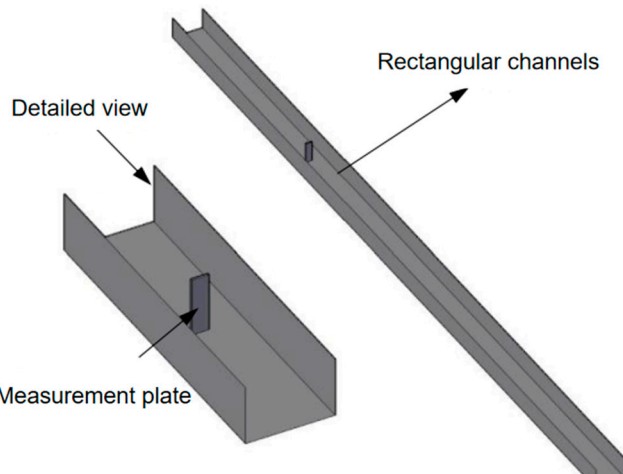

**Figure 2.** Layout diagram of the geometric model.

### 2.3. Computational Domain Meshing

In the numerical calculation process, the size and number of grids affect the accuracy of the results. Therefore, the independence of the grid is checked by monitoring the changes in water level upstream and the velocity of the measuring plate under different grid sizes. We carried out a simulation for different mesh sizes of 20, 15, 10, 8, 5, 2.5 and 1.0 mm. The results are shown in Figure 3. It can be seen that the discrepancy between the simulation results for the mesh sizes of 1 and 10 mm is only 0.3%. We, therefore, conclude that mesh size is not the main factor affecting the simulation results at this time when there is a strong correspondence between the simulation results and the physical test results. Because a mesh size below 10 mm was suitable for the study purposes, this was the size selected for our simulation.

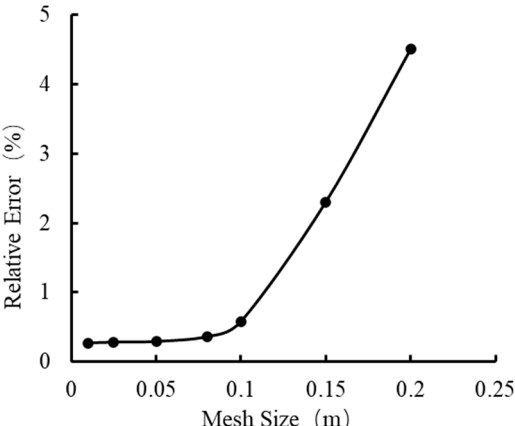

**Figure 3.** Mesh sensitivity analysis.

The model was imported into ICEM CFD to mesh the computational domain. In order to ensure calculation accuracy, improve the meshing quality, and minimize the calculation time, a minimum size of 2.5 mm and a maximum size of 10 mm were used for meshing the model. Because of the simple structure of the measurement plate and the rectangular channel, the structured mesh was used to divide the model. In the mesh division, considering that the area near the measurement plate is the main area of study, the mesh near the upstream and downstream was encrypted, and a mesh size of 2.5 mm was chosen, while the mesh in the area far from the measurement plate was chosen as 10 mm size. In order to ensure the two parts of the mesh are more smoothly connected, except for the mesh size near the plate surface that was 2.5 mm, the mesh size of other parts of the encrypted area was 5 mm. The total mesh number after mesh division was about 710,000, and the mesh division results are shown in Figure 4.

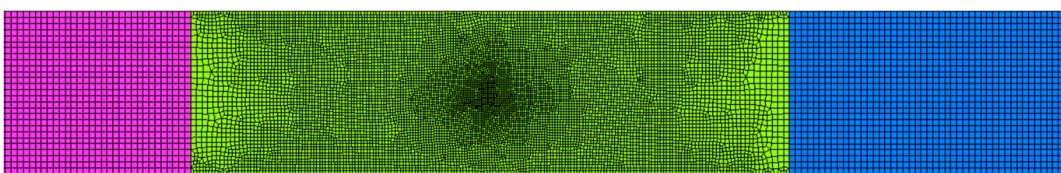

**Figure 4.** Diagram of mesh division.

### 2.4. Computational Model and Boundary Conditions

We used Fluent software for model simulations and calculations. In choosing the computational model, since the RNG $k$-$\varepsilon$ turbulence model is derived by using the mathematical method of reformed group and compared with the standard $k$-$\varepsilon$ model, the term is added to the equation of turbulent kinetic energy $k$ and dissipation rate $\varepsilon$, which makes its calculation of turbulence more accurate and efficient [25]. Therefore, the RNG $k$-$\varepsilon$ turbulence model

was chosen for the simulation study in this paper. The equations of turbulent kinetic energy $k$ and dissipation rate $\varepsilon$ for this model are [26]:

$$\rho \frac{dk}{dt} = \frac{\partial}{\partial x_i} \left[ \left( a_k \mu_{eff} \right) \frac{\partial k}{\partial x_i} \right] + G_k + G_b - \rho \varepsilon - Y_M \tag{3}$$

$$\rho \frac{d\varepsilon}{dt} = \frac{\partial}{\partial x_i} \left[ \left( a_\varepsilon \mu_{eff} \right) \frac{\partial \varepsilon}{\partial x_i} \right] + C_{1\varepsilon} \frac{\varepsilon}{k} (G_k + C_{3\varepsilon} G_b) - C_{2\varepsilon} \rho \frac{\varepsilon^2}{k} - R \tag{4}$$

In the formula, $k$ is the turbulent kinetic energy, $m^2/s^2$; $\varepsilon$ is the turbulent energy dissipation rate, $kg \cdot m^2/s^3$; $G_k$ denotes the turbulent energy generation due to the mean velocity gradient; $G_b$ is the turbulent energy generation due to the buoyancy effect; $Y_M$ is the effect of the pulsating expansion of the compressible velocity turbulence on the total dissipation rate; $\alpha_k$ and $\alpha_\varepsilon$ are the inverse of the effective turbulent Prandtl number of the turbulent kinetic energy $k$ and the dissipation rate $\varepsilon$; $\mu_{eff}$ is the effective kinetic viscosity coefficient of the fluid, which is equal to the sum of $\mu$ and $\mu_t$, $\mu_t = \rho C_\mu \frac{k^2}{\varepsilon}$, $C_\mu = 0.0845$. As default constants, $C_{1\varepsilon} = 1.44$, $C_{2\varepsilon} = 1.92$, $C_{3\varepsilon} = 0.09$, and the turbulent Prandtl number of the turbulent kinetic energy $k$ and dissipation rate $\varepsilon$ are $\sigma_k = 1.0$ and $\sigma_\varepsilon = 1.3$, respectively.

This simulation uses transient simulation, where the initial state in the entire flow field is all air, and after the calculation starts, the channel is gradually filled with water as time advances. The VOF model is used for the free surface of the open channel flow, which is an exterior tracking method built under a fixed Eulerian mesh and can be used when one or more mutually incompatible fluid interfaces are required. In the VOF model, different fluid components share a set of momentum equations, and by introducing the variable of phase volume fraction, the tracking of the phase interface of each computational cell is realized, so as to achieve the purpose of simulating multiple unmixed fluids.

In setting the boundary conditions, mainly the inlet boundary conditions and outlet boundary conditions are set. As shown in Table 1. When setting the inlet boundary conditions, the upstream inlet of the channel is divided into water inlet and air inlet according to the characteristics of the channel model, where the air inlet adopts the pressure inlet, the water volume fraction is set to 0, and the reference pressure is atmospheric pressure; the water inlet adopts the velocity inlet, and the water volume fraction is set to 1; the upper surface of the channel is set as the air pressure inlet, and the reference pressure is atmospheric pressure. The outlet boundary is set as the pressure outlet, and the reference pressure is atmospheric pressure.

**Table 1.** Boundary conditions for numerical simulation.

| Geometry | Boundary Conditions |
|---|---|
| Water inlet | Velocity inlet |
| Air inlet | Pressure inlet |
| Outlet | Pressure outlet |
| Top boundary | Pressure inlet |
| Others | No-slip wall |

Numerical simulations were carried out at the flow rates of 20 to 105 $m^3/h$. By monitoring the simulation results of the flow rate at the outlet boundary and the force on the measurement plate, it was found that after about 75 s from the initial time, the mass flux difference between the velocity inlet and the pressure outlet is within 1%, and the force on the measurement plate is almost constant, at which time the model simulation could reach a stable state. In order to ensure the stability of the simulation results, the calculation time was set to 105 s.

## 3. Experimental Verification of Simulation Results

In order to verify the reliability of the simulation results of the mathematical model selected for the flow measurement process of the spring-plate flow measurement device, it is necessary to conduct a flow measurement test using the spring-plate flow measurement device and to compare the simulation and the experimental results. The layout of the test system is shown in Figure 5.

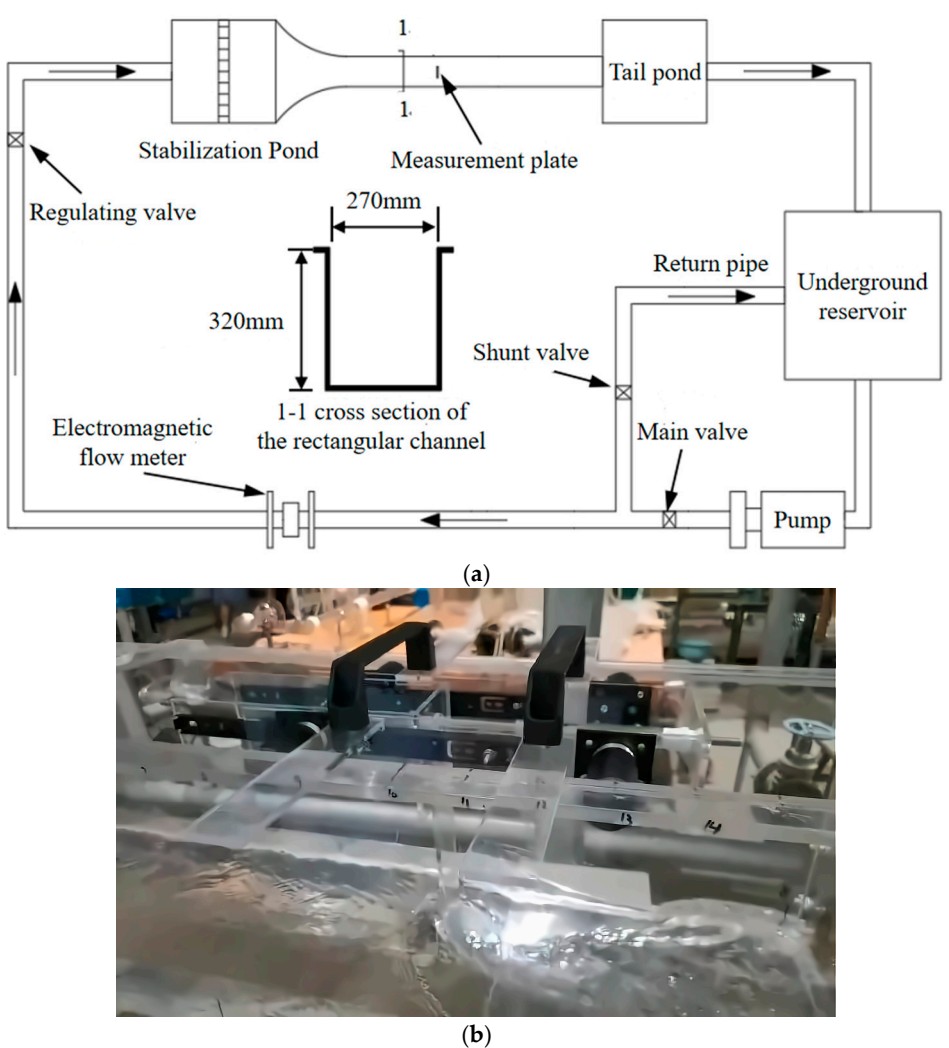

(**a**)

(**b**)

**Figure 5.** Layout of test system: (**a**) model diagram of the test system; (**b**) use demonstration of the spring-plate flow measurement device.

The test system mainly consisted of an underground reservoir, water pump, pipeline, electromagnetic flow meter, water tank, water stabilization plate, and rectangular channel. Water was pumped from the underground reservoir and transported to the upstream tank through the pipeline. The electromagnetic flow meter was installed on the pipeline, the flow rate of the water conveyed by the pipeline could be read through the display panel in real time, and the size of the flow rate could be adjusted by turning the valve. The water flow into the upstream tank was stabilized and then conveyed to the rectangular channel through the trumpet. The water in the channel was conveyed to the downstream tank and discharged back to the underground reservoir by the drainage outlet, forming a complete underground reservoir circulation system. The width of the rectangular channel was 270 mm, the height was 320 mm, and the bottom slope was 1/1000. The wire diameter of the compression spring used in the test was 1 mm, the middle diameter was 20 mm, and the stiffness coefficient was 0.1195 N/mm. The shortened length could be read by the scale

on the measurement sleeve, which can be accurate to 1 mm. The thickness of the selected measurement plate was 5 mm, and the widths of plate *B* were 30, 40, 50, and 60 mm, which is consistent with the simulated dimensions.

In this experiment, the force on the plate surface in the channel and the water level 0.2 m upstream of the measurement plate were measured under the conditions of 20–80 m$^3$/h flow, and the experimental results were compared with the simulated values. As shown in Figure 6, the main operation steps of the experiment were as follows:

(1) The spring-plate flow measurement device was fixed on the rectangular channel, and forces of different sizes were applied to the measurement plate through the pressure sensor. The size of the applied force was read, and the distance to which the spring was compressed was recorded using the scale on the measurement sleeve. The frictional force $f'$ between the devices and the stiffness coefficient of the springs can be obtained from the relationship between them.

(2) The pump was turned on to allow water transmission, and the flow rate in the channel was adjusted to 20 m$^3$/h via the electromagnetic flow meter and the valve. The distance to which the spring was compressed was recorded after the flow rate was stabilized, and the average water level was measured 0.2 m upstream of the measurement plate.

(3) The flow rate was adjusted and measured at 10 m$^3$/h intervals of flow rate in the range of 20–80 m$^3$/h, and the distance to which the spring was compressed and the average water level were recorded 0.2 m upstream of the measurement plate. The flow rate was then adjusted back to 20 m$^3$/h, and the above steps were repeated three times for each flow rate. After the measurement was completed, the results of the three measurements were averaged.

(4) The spring force $F_k$ could be calculated using the distance to which the spring was compressed and the spring stiffness coefficient, and the measured friction force $f'$ between the devices was then added to obtain the force on the surface of the measurement plate.

(5) The results of the experiment were compared with the simulated values in the graph, and the relative error between the two was calculated.

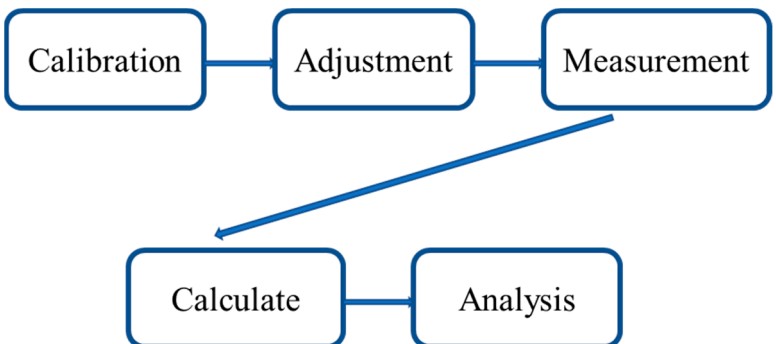

**Figure 6.** Test process.

When the flow moves in the channel, the flow pattern changes due to the constraints of channel and plate. Comparing and observing the numerical simulation results with the flow pattern changes during the experiment can very well verify the authenticity of the simulation results. As can be seen in Figure 5b, the upstream water level of the spring-plate type flow meter is higher than the downstream water level. The upstream flow pattern is gentle, while the downstream flow pattern is disordered and the water surface fluctuates uncertainly. Near the measuring plate, the plate shrinks the channel section, impeding the flow downstream and reducing the flow rate. It makes the water level higher and the kinetic energy is converted into potential energy. When the water flows around the measuring plate, the water level on both sides suddenly drops and the flow velocity

increases. Potential energy is converted into kinetic energy. Furthermore, some of the flow in this area flows back to the downstream surface of the measuring plate. Therefore, the measuring plate located in the center of the channel hinders the normal flow of water, but this blocking effect weakens away from the measuring plate. The water flow on both sides of the canal walls flows against the wall due to inertia, and the water level drops more slowly compared to the water level near the measuring plate. The water flow on both sides intersects at the centerline downstream of the measuring plate, with a high flow velocity, forming a hydraulic jump that connects with the downstream water surface, resulting in a "basin-like" water surface flow pattern at the downstream.

Figure 7 shows the comparison of the force on the surface of the spring-plate flow measurement device under simulated and experimental flow measurement conditions for four types of plate widths. From the figure, it can be seen that when the plate width or flow rate increases, the force on the surface of the measurement plate will also become larger. The simulated value of the force on the plate surface basically matches the experimental value. The average relative errors between the simulated and experimental values are 3.43%, 5.85%, 4.32%, and 3.94% for plate widths of 30, 40, 50, and 60 mm, respectively, and the relative errors are less than 6.26%. The simulated results of the force on the plate surface are consistent with the reality.

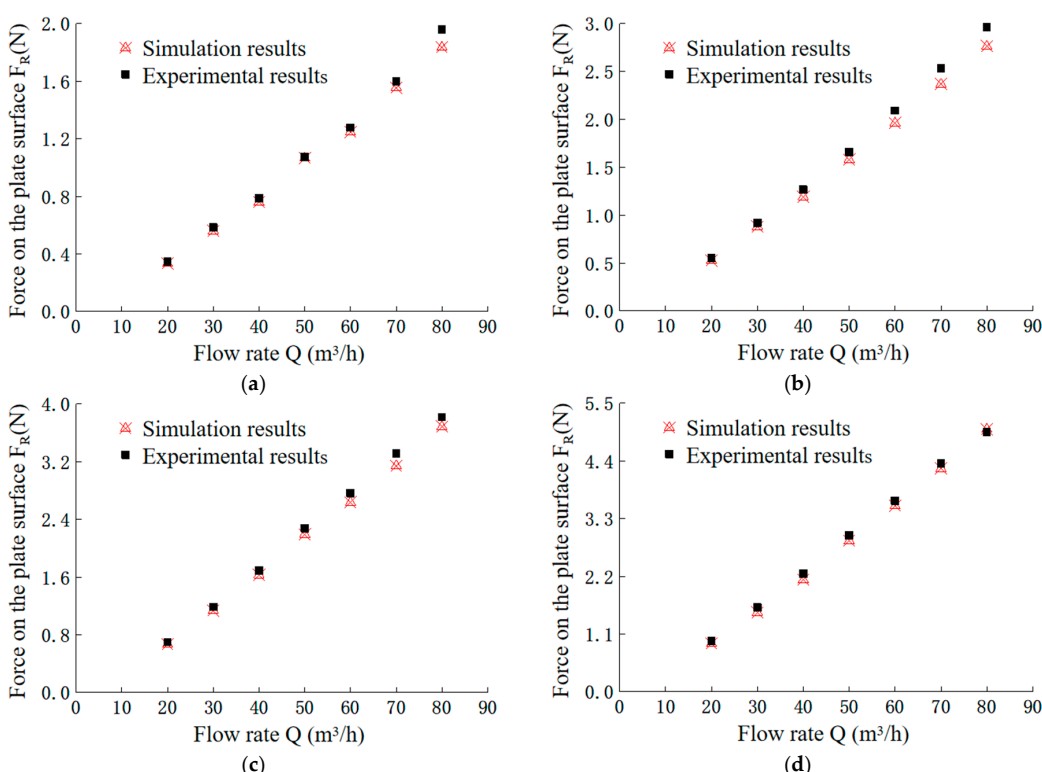

**Figure 7.** Comparison between the experimental and simulation results of the force on measurement plate surface: (**a**) *B* = 30 mm; (**b**) *B* = 40 mm; (**c**) *B* = 50 mm; (**d**) *B* = 60 mm.

Figure 8 shows the comparison between the numerical simulation results and the measured experimental results of the average water depth 0.2 m upstream of the measurement plate for different flow rates during the flow measurement of the spring-plate flow measurement device. It can be seen from the figure that when the plate width or flow rate increases, the depth of water upstream of the plate will also become larger. The simulated value of upstream water depth basically matches the experimental value. The average relative error of the upstream water level was calculated as 1.77%, 1.96%, 3.00%, and 1.72% for plate widths of 30, 40, 50, and 60 mm, respectively, and the relative error

value of upstream water level was less than 5%. The simulation results of water depth are consistent with the reality.

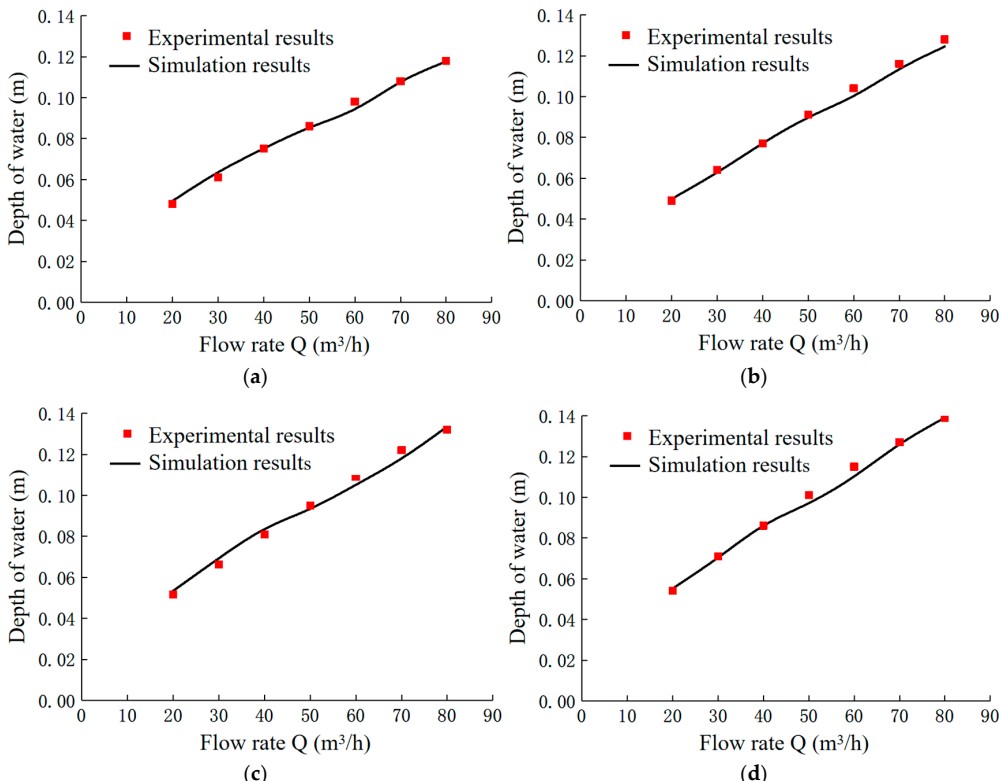

**Figure 8.** Comparison between the experimental and simulation results of average water depths at 0.2 m upstream of the measurement plate: (**a**) B = 30 mm; (**b**) B = 40 mm; (**c**) B = 50 mm; (**d**) B = 60 mm.

In summary, the simulated values of the spring-plate flow measurement device do not differ much from the measured values under the simulated and experimental flow measurement conditions, which shows that simulating the open channel spring-plate flow measurement device using Fluent software is feasible.

## 4. Results and Analysis

### 4.1. Relationship between the Flow Rate and the Force on the Plate

Through theoretical analysis, it can be concluded that the main factors affecting the measurement of flow $Q$ by the spring plate flow-measuring device are: gravity acceleration $g$, hydraulic radius R of the cross section, density of water $\rho_w$, dynamic viscosity coefficient of water $v$, force of water flow on the measuring plate $F_R$ ($F_R = F_K + f'$), width B of the measuring plate, thickness of the measuring plate $\delta$, roughness of the channel $n$, and slope of the channel $i$. Based on the $\pi$ theorem and the above influencing factors, the functional relationship is: $f(Q, g, R, \rho_w, v, F_R, B, \delta, n, i) = 0$. Based on dimensional analysis, the dimensionless equation is obtained as follows: $\varphi(\pi_1, \pi_2, \pi_3, \pi_4, \pi_5, \pi_6, \pi_7) = 0$. Here, $F_R$, $\rho_w$, $g$ are selected as the basic physical quantities for research. Then, dimensionless $\pi$ satisfies the equation:

$$\pi_k = \frac{x_k}{F_R^{a_k} \rho_w^{b_k} g^{c_k}}$$

The equation is written as:

$$\varphi\left(\frac{Q}{F_R^{\frac{5}{6}} \rho_w^{-\frac{5}{6}} g^{-\frac{1}{3}}}, \frac{R}{F_R^{\frac{1}{3}} \rho_w^{-\frac{1}{3}} g^{-\frac{1}{3}}}, \frac{v}{F_R^{\frac{1}{2}} \rho_w^{\frac{1}{2}}}, \frac{B}{F_R^{\frac{1}{3}} \rho_w^{-\frac{1}{3}} g^{-\frac{1}{3}}}, \frac{\delta}{F_R^{\frac{1}{3}} \rho_w^{-\frac{1}{3}} g^{-\frac{1}{3}}}, n, i\right) = 0$$

It can be derived that:

$$\frac{Q}{F_R^{\frac{5}{6}}\rho_w^{-\frac{5}{6}}g^{-\frac{1}{3}}} = \varphi\left(\frac{R}{F_R^{\frac{1}{3}}\rho_w^{-\frac{1}{3}}g^{-\frac{1}{3}}}, \frac{v}{F_R^{\frac{1}{2}}\rho_w^{\frac{1}{2}}}, \frac{B}{F_R^{\frac{1}{3}}\rho_w^{-\frac{1}{3}}g^{-\frac{1}{3}}}, \frac{\delta}{F_R^{\frac{1}{3}}\rho_w^{-\frac{1}{3}}g^{-\frac{1}{3}}}, n, i\right)$$

Then,

$$Q = \left(\frac{F_R}{\rho_w g^{\frac{2}{5}}}\right)^{\frac{5}{6}}\varphi\left(\frac{R}{F_R^{\frac{1}{3}}\rho_w^{-\frac{1}{3}}g^{-\frac{1}{3}}}, \frac{v}{F_R^{\frac{1}{2}}\rho_w^{\frac{1}{2}}}, \frac{B}{F_R^{\frac{1}{3}}\rho_w^{-\frac{1}{3}}g^{-\frac{1}{3}}}, \frac{\delta}{F_R^{\frac{1}{3}}\rho_w^{-\frac{1}{3}}g^{-\frac{1}{3}}}, n, i\right)$$

The parameters are defined as:

$$C = \left(\frac{1}{\rho_w g^{\frac{2}{5}}}\right)^{\frac{5}{6}}\varphi\left(\frac{R}{F_R^{\frac{1}{3}}\rho_w^{-\frac{1}{3}}g^{-\frac{1}{3}}}, \frac{v}{F_R^{\frac{1}{2}}\rho_w^{\frac{1}{2}}}, \frac{B}{F_R^{\frac{1}{3}}\rho_w^{-\frac{1}{3}}g^{-\frac{1}{3}}}, \frac{\delta}{F_R^{\frac{1}{3}}\rho_w^{-\frac{1}{3}}g^{-\frac{1}{3}}}, n, i\right)$$

The equation can be represented as:

$$Q = C \cdot F_R^{5/6} \tag{5}$$

It can be initially obtained using the method of dimensional analysis [19]. In the formula, $Q$ is the flow rate of the channel cross section; $C$ is the coefficient related to the hydraulic radius of the cross section, the dynamic viscosity coefficient of the water, the width of the measurement plate, and the slope of the channel, which can be obtained by experimental fitting; $F_R$ is the force on the surface of the measurement plate. Through numerical simulation, the force on the plate surface $F_R$ and the flow rate $Q$ in the channel can be derived, and Matlab is used to fit the two. The fitting results are shown in Figure 9, and the specific formulas are shown in Table 2.

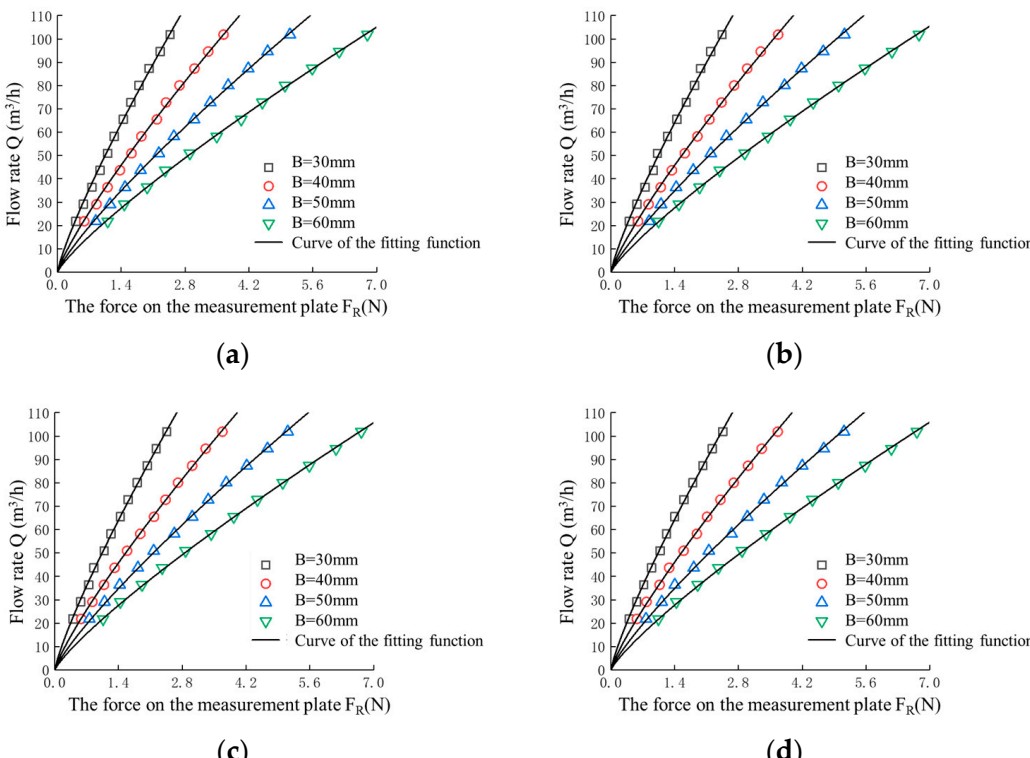

**Figure 9.** *Cont.*

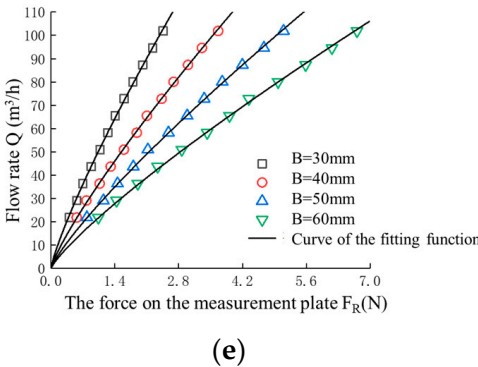

(**e**)

**Figure 9.** The relationship between the force on the measurement plate and the flow rate of the spring-plate flow measurement device: (**a**) $i = 1/500$; (**b**) $i = 1/1000$; (**c**) $i = 1/2500$; (**d**) $i = 1/5000$; (**e**) $i = 1/10{,}000$.

**Table 2.** Fitting relationship between simulated flow rate and plate force of spring-plate flow measurement device under different working conditions.

| Width | Slope | Formula | $R^2$ | Root-Mean-Square Error |
|---|---|---|---|---|
| 30 mm | 1/500 | $Q = 48.15 \cdot F_R^{\frac{5}{6}}$ | 0.9978 | 1.2220 |
| | 1/1000 | $Q = 48.46 \cdot F_R^{\frac{5}{6}}$ | 0.9974 | 1.3280 |
| | 1/2500 | $Q = 48.40 \cdot F_R^{\frac{5}{6}}$ | 0.9989 | 0.8774 |
| | 1/5000 | $Q = 48.57 \cdot F_R^{\frac{5}{6}}$ | 0.9988 | 0.8931 |
| | 1/10,000 | $Q = 48.55 \cdot F_R^{\frac{5}{6}}$ | 0.9992 | 0.7537 |
| 40 mm | 1/500 | $Q = 34.71 \cdot F_R^{\frac{5}{6}}$ | 0.9984 | 1.0340 |
| | 1/1000 | $Q = 34.62 \cdot F_R^{\frac{5}{6}}$ | 0.9991 | 0.7907 |
| | 1/2500 | $Q = 34.62 \cdot F_R^{\frac{5}{6}}$ | 0.9996 | 0.5114 |
| | 1/5000 | $Q = 34.81 \cdot F_R^{\frac{5}{6}}$ | 0.9995 | 0.6035 |
| | 1/10,000 | $Q = 34.79 \cdot F_R^{\frac{5}{6}}$ | 0.9995 | 0.5933 |
| 50 mm | 1/500 | $Q = 26.40 \cdot F_R^{\frac{5}{6}}$ | 0.9995 | 0.5691 |
| | 1/1000 | $Q = 26.37 \cdot F_R^{\frac{5}{6}}$ | 0.9993 | 0.7004 |
| | 1/2500 | $Q = 26.32 \cdot F_R^{\frac{5}{6}}$ | 0.9994 | 0.6578 |
| | 1/5000 | $Q = 26.35 \cdot F_R^{\frac{5}{6}}$ | 0.9991 | 0.8043 |
| | 1/10,000 | $Q = 26.40 \cdot F_R^{\frac{5}{6}}$ | 0.9994 | 0.6232 |
| 60 mm | 1/500 | $Q = 20.76 \cdot F_R^{\frac{5}{6}}$ | 0.9994 | 0.6518 |
| | 1/1000 | $Q = 20.84 \cdot F_R^{\frac{5}{6}}$ | 0.9998 | 0.3799 |
| | 1/2500 | $Q = 20.89 \cdot F_R^{\frac{5}{6}}$ | 0.9996 | 0.5290 |
| | 1/5000 | $Q = 20.93 \cdot F_R^{\frac{5}{6}}$ | 0.9996 | 0.5146 |
| | 1/10,000 | $Q = 20.98 \cdot F_R^{\frac{5}{6}}$ | 0.9992 | 0.7370 |

The following are the conclusions drawn from Figure 9 and Table 2. For a certain channel slope, the force on the measurement plates of different widths increases with the increase in the flow rate in the channel section. The flow rate $Q$ in the rectangular channel has a good linear correlation with the 5th/6th power of the plate force $F_R$, and the residual $R^2$ of the fitting results are above 0.99, which shows that the equation obtained by

the method of dimensional analysis to describe the relationship between the flow rate $Q$ in the rectangular channel and the plate surface force $F_R$ is feasible. When the width of the measurement plate changes, the coefficient C of the fitting formula also changes. The table shows that the overall root-mean-square error of the fitted formula tends to increase when the plate width decreases. It can be seen that when using the spring-plate flow measurement device to measure flow in the same channel, the variation in the plate width of the measurement plate will have a certain impact on the accuracy of the flow measurement.

For the flow measurement of spring plate flow-measuring device in a rectangular channel, the following formula and reference table are used for calculation. Table 3 shows the coefficients of common slopes and plate widths in practice.

$$Q = C \cdot F_R^{5/6} \tag{6}$$

**Table 3.** C value under common conditions in practice.

| Slope | | 1/500 | 1/1000 | 1/2500 | 1/5000 | 1/10,000 |
|---|---|---|---|---|---|---|
| | 30 mm | 48.15 | 48.46 | 48.40 | 48.57 | 48.55 |
| | 40 mm | 34.71 | 34.62 | 34.62 | 34.81 | 34.79 |
| Width | 50 mm | 26.40 | 26.37 | 26.32 | 26.35 | 26.40 |
| | 60 mm | 20.76 | 20.84 | 20.89 | 20.93 | 20.98 |

In order to obtain the measurement accuracy of the spring-plate flow measurement device, error analysis of its fitted flow measurement equation under different plate width conditions is performed. The flow measurement error is calculated as follows:

$$E = (Q_c - Q_m)/Q_m \times 100\% \tag{7}$$

In the formula, $e$ is the relative error of the flow measurement result, $Q_c$ is the flow rate calculated by fitting the formula, and $Q_m$ is the flow rate derived from numerical simulations. The results show that the maximum relative errors of the channel flow rate calculated using the corresponding fitting equations in Table 2 and Equation (6) compared with the simulated actual flow rate were 5.65% for different measurement plate widths, and the average relative errors of each formula did no exceed 2.56%. This shows that the accuracy of channel flow measurement is high when using the spring-plate flow measurement devices.

### 4.2. Distribution of the Pressure on the Surface of the Measurement Plate

The spring-plate flow measurement device relies on the pressure difference between the upstream and downstream plates of the measurement plate to push the measurement guide to squeeze the spring to produce deformation in order to calculate the channel flow. In order to investigate the hydraulic characteristics near the spring-plate flow measurement device, the static pressure distribution on the upstream and downstream plate surface of the measurement plate was analyzed. Under the condition that the flow rate in the channel is 50 m$^3$/h, the static pressure distribution on the upstream and downstream plate surfaces of four different plate widths were analyzed, as shown in Figure 10. In the figure, the pressure distribution on the upstream surface of the measurement plate is shown on the left, and the pressure distribution on the downstream surface of the measurement plate is shown on the right.

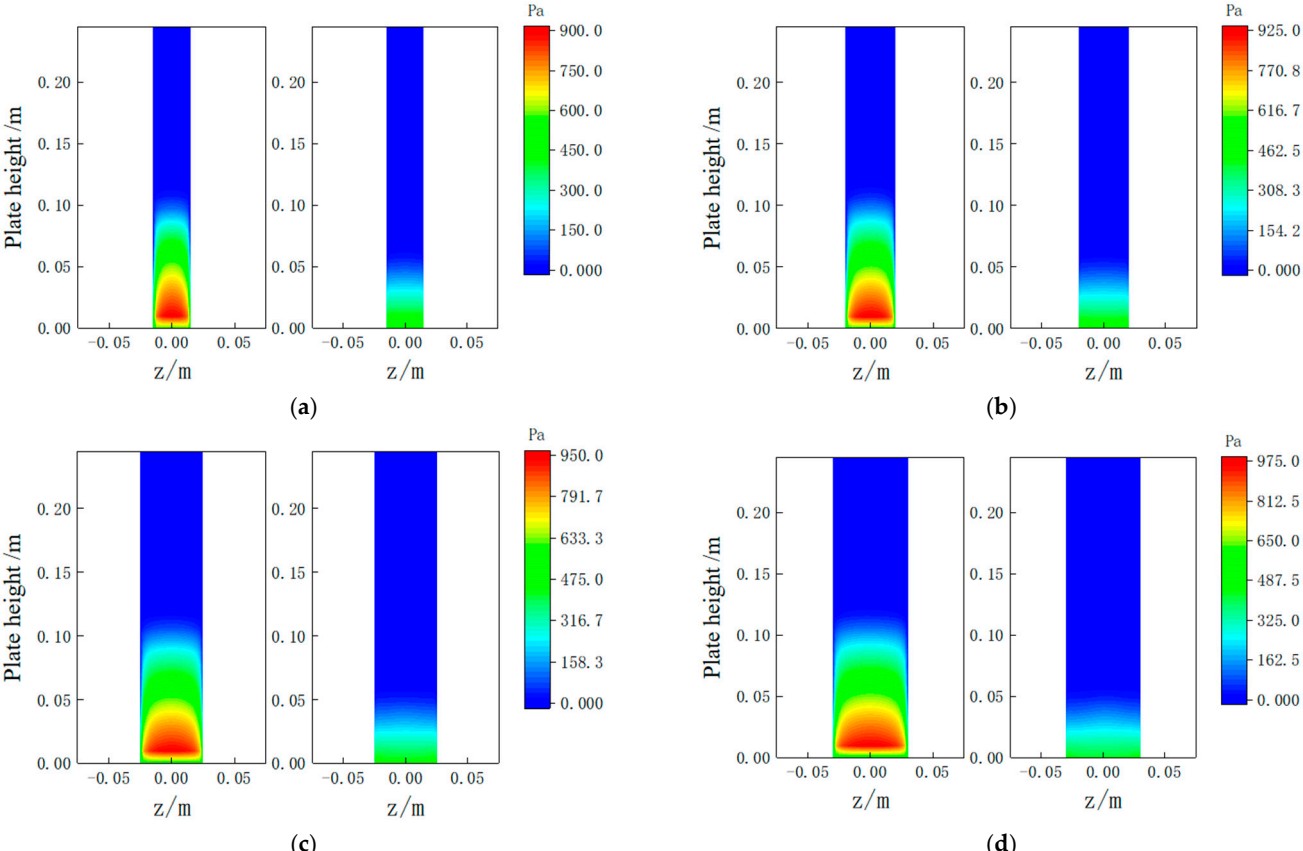

**Figure 10.** Distribution of the pressure on the surface of the measurement plate: (**a**) *B* = 30 mm; (**b**) *B* = 40 mm; (**c**) *B* = 50 mm; (**d**) *B* = 60 mm.

As can be seen from Figure 10, due to the stable flow pattern upstream of the measurement plate, the pressure change law for water flow on the measurement plate is similar to that of static pressure change, generally and gradually increasing along the direction of water depth. When the water flows through the measurement plate, winding flow occurs at the measurement plate, and when the water reaches the front of the measurement plate, the kinetic energy of the water is converted into potential energy, the upstream water level is raised, and the water level is higher in the middle of the measurement plate than the at the edges. At the same time, the water level on both sides of the measurement plate falls, the potential energy is converted into kinetic energy, and the flow velocity increases. From Bernoulli's equation, the mechanical energy of a fluid mass in a flow line is conserved, and the pressure decreases as the kinetic energy of the fluid increases. In summary, at the same horizontal height, the pressure of water flow is higher in the middle of the measurement plate than at the edge.

Comparing the pressure distribution upstream and downstream of the plate under different plate widths, it can be found that as the plate width increases, the cross-sectional area of the channel decreases, which leads to a higher water level at the upstream of the plate, and thus the pressure of water acting on the plate simultaneously increases.

### 4.3. Velocity Distribution

In order to further investigate the hydraulic characteristics of the spring-plate flow measurement device, the velocity distribution of the flow in the channel during the flow measurement process was analyzed. The flow velocity distribution of the surface and some of the characteristic sections of the channel were studied under the conditions of 60 mm plate width and *Q* = 40 m$^3$/h.

Figure 11 shows the flow velocity distribution on the surface of the water in the channel during flow measurement of the spring-plate flow measurement device. It can be seen that the upstream velocity distribution of the spring-plate flow measurement device is relatively uniform, and the velocity is small. As it gets closer to the plate, the flow velocity decreases at the middle line of the channel, and the velocity decreases to 0 at the front of the plate. Water on both sides of the flow separates from the measurement plate as it passes through the plate, and the velocity increases continuously and reaches a maximum at a certain distance after the measurement plate. The water flow through the measurement plate is diffused downstream of the plate, and the water flows on both sides intersect each other, forming a hydraulic jump. After the jump, the water flows continuously intersect and collide with the diffused water flow, disturbing the water flow pattern, which gradually stabilizes downstream and is restored to slow flow. There is a backflow area near the downstream surface of the measurement plate, and the flow velocity is small, close to 0.

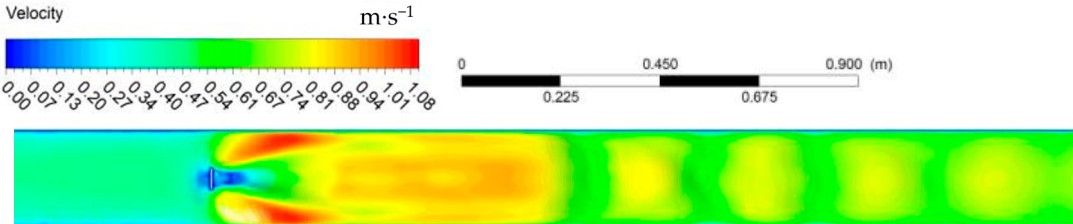

**Figure 11.** Surface velocity distribution of the water in the channel.

Four characteristic sections were selected in the rectangular channel to analyze its flow velocity distribution, and the selected sections are shown in Figure 12. The flow velocity distribution of the characteristic cross-sections is shown in Figure 13. From Figure 13, it can be seen that:

(1) Section 1-1 is located at the stable water surface upstream of the measurement plate, the flow velocity is evenly distributed, the flow velocity near the side wall of the channel decreases sharply, and the flow velocity on the side wall is 0. The maximum flow velocity of the channel section is located below the liquid surface, and the flow velocity at the gas–liquid junction is relatively small.

(2) Section 2-2 is located upstream of the measurement plate and close to the plate, and the flow velocity near the middle line of the channel is relatively low, at about 0.45 m/s. The closer to the sides of the channel, the larger the velocity, and the maximum value is about 0.58 m/s. The flow at the middle line is obstructed by the measurement plate, the water surface is congested upward, and the flow velocity is reduced. The flow velocity at the water surface is about 0.28 m/s.

(3) Section 3-3 is close to the downstream end of the measurement plate, and at this time, the water flow through the measurement plate here forms a backflow area, and the flow velocity at the middle line is close to 0. The flow velocity from the middle line to both sides of the channel increases at first and then decreases. The flow velocity reaches its maximum of about 0.95 m/s at the position near the edge of both sides of the measurement plate. The flow velocity decreases sharply near the side wall of the channel.

(4) Section 4-4 is a downstream position away from the measurement plate. The flow velocity distribution is more uniform, the flow velocity at the side wall is low, and the flow velocity near the middle line of the channel is higher. At this time, the measurement plate has little influence on the flow pattern, and the water is in a slow flow state.

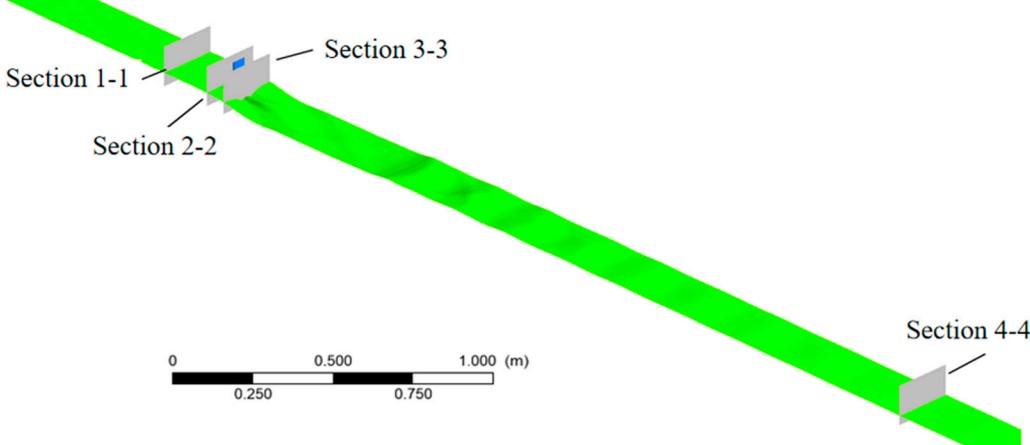

**Figure 12.** Distribution of the sections along the route.

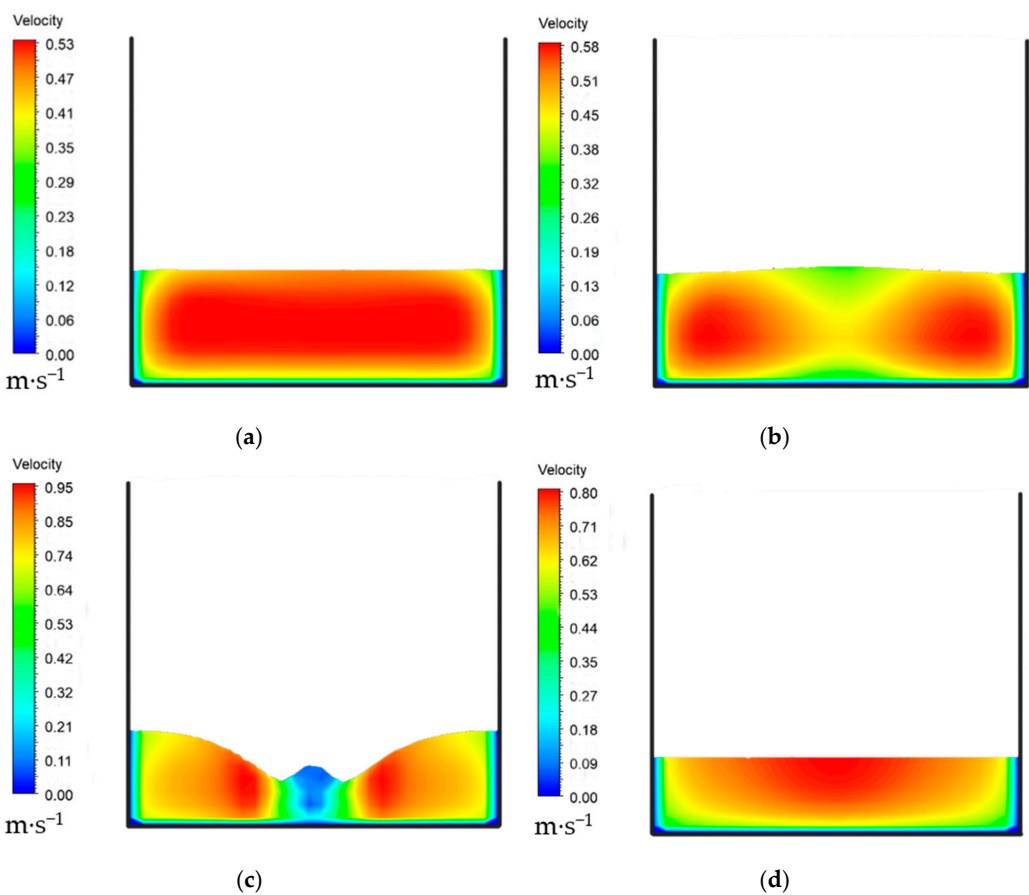

**Figure 13.** Distribution of the flow velocity at characteristic sections in the channel: (**a**) Section 1-1; (**b**) Section 2-2; (**c**) Section 3-3; (**d**) Section 4-4.

*4.4. Head Loss*

When using the spring-plate flow measurement device to measure flow, it is necessary to extend the measurement plate into the water, which will cause obstruction to the flow of water and thus generate head loss. In order to reduce the head loss caused by the use of the spring-plate flow measurement devices, the measurement plate needs to be set to an appropriate depth, so the head loss caused by the spring-plate flow measurement device with different measurement plate widths was analyzed here. The control sections a–a and

b–b were respectively selected at the stable water surface upstream and downstream of the spring-plate flow measurement device, and according to the energy equation, the equation of head loss is as follows:

$$h_j = \left( h_1 + \frac{a_1 v_1^2}{2g} \right) - \left( h_2 + \frac{a_2 v_2^2}{2g} \right) \tag{8}$$

In the formula, $h_j$ is the head loss, $h_1$ is the depth of water at the a–a section, $h_2$ is the depth of water at the b–b section, $v_1$ is the average flow velocity at the a–a section, $v_2$ is the average flow velocity at the b–b section, and $a$ is the kinetic energy correction factor. In order to better reflect the impact of the spring-plate flow measurement device on head loss, the percentage $\zeta$ between the head loss generated by devices with different plate widths and the total head in the upstream was analyzed, that is:

$$\zeta = \frac{h_j}{\left( h_1 + \frac{a_1 v_1^2}{2g} \right)} \tag{9}$$

Figure 14 shows the percentage of head loss in the upstream total head generated by the spring-plate flow measurement device with different plate widths during the flow measurement in the channel. As can be seen from the figure, the head loss ratio was within 10% when the width of the plate was 30 and 40 mm, and above 10% when the width of the plate was 50 and 60 mm. For the same flow rate, the head loss ratio increased with the increase in the measurement plate width. This is because when the width of the measurement plate is small, the lateral contraction of the rectangular channel is smaller, the impact on the flow of water through the measurement plate is smaller, and the head loss ratio is also smaller. When the width of the measurement plate is large, the lateral contraction of the measurement plate on the channel is larger, the obstruction of the water flow is enhanced, intensifying the turbulence inside the water flow, energy consumption increases, and the head loss ratio will also increase. Therefore, in the actual use of the spring-plate flow measurement device, it is necessary to select an appropriate plate width for the measurement plate in order to reduce head loss.

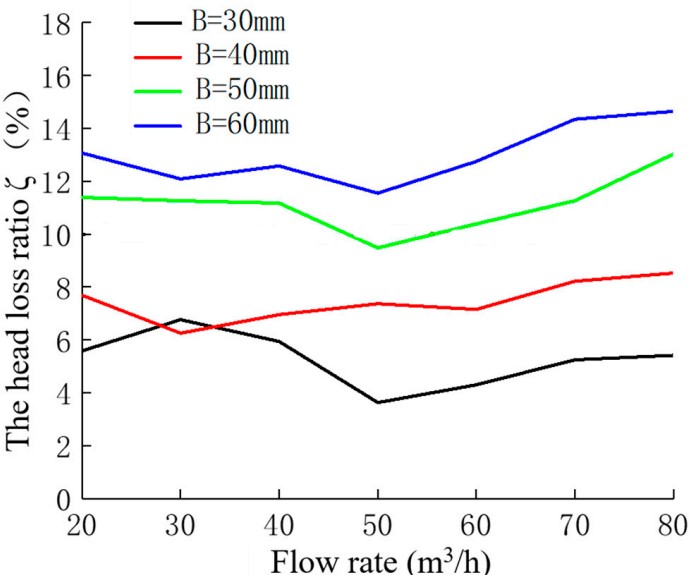

**Figure 14.** Head loss ratios with different plate widths.

## 5. Discussion

We designed a spring plate flow measurement device with different plate widths to meet its applicability in channels of irrigation areas. Compared with other flow measurement devices, the spring-plate flow measurement device has the advantages of simple structure, portability, and low price. Through the study of this paper, it was found that its measurement accuracy is high and the head loss generated when used is small. However, since the flow rate is derived by bringing the force of the water flow on the measurement plate into the equation, the coefficients of the calculation equation will vary under different channel conditions, so the influence of various parameters of the channel, such as the channel bottom slope and channel width, on the coefficients of the calculation equation needs to be derived in subsequent experiments. In addition, the flow rate used in this study was 20–105 m$^3$/h. Further studies are needed to determine whether the higher flow rate would affect the accuracy of the measurement results. Furthermore, to advance intelligent water measurement and enhance the automation of the agricultural industry, future improvements to the spring plate flow-measuring device studied in this paper may involve equipping it with various sensors to facilitate more efficient and rapid measurements.

## 6. Conclusions

1. In the process of using a spring-plate flow measurement device to measure flow, the plate force increases with the increasing flow rate in the channel, and the plate force is related to the flow rate in the channel to the power of 5/6; furthermore, the coefficient C will change with the variation in measurement plate parameters. The results of the numerical simulation are fitted to derive the coefficient C at different measurement plate width and slope conditions.
2. The relative error between the simulated value and the calculated value of the fitted equation from the spring-plate flow measurement device is small, and the measurement accuracy of the channel flow rate in the flow measurement process is high. The width of the measurement plate has a certain influence on the measurement error; the smaller the plate width, the larger the error of the measurement.
3. The pressure change law for water flow on the measuring plate is similar to the distribution of static pressure, generally and gradually increasing along with the direction of the water depth. At the same horizontal height, the pressure of water flow is higher in the middle of the measurement plate than that at the edge. When the width of the measurement plate increases, the pressure of the water acting on the plate will also increase.
4. When using the spring-plate flow measurement device to measure flow, the upstream flow velocity is small, and the water level increases due to obstruction of the plate. The water level then falls to connect with the downstream water surface, and the flow velocity increases. The overall velocity of water flow shows distribution according to the law where it is low near the middle line of the channel and increases and then decreases from the middle line to both sides of the channel, and the velocity reaches its maximum at the position near the edge of both sides of the measurement plate.
5. Using the spring-plate flow measurement device to measure the flow will produce head loss. When the measurement plate width does not exceed 40 mm, the ratio of head loss is less than 10%. When the measurement plate width is not less than 50 mm, the ratio of head loss is more than 10%. The head loss increases with the increase in the width of the measurement plate.
6. When the width of the measurement plate is 40 mm, the head loss ratio caused by the channel water flow is less than 10%, and the error of measuring the flow is low, which can ensure the accuracy and range requirements at the same time, so the measurement plate width of 40 mm is recommended.

**Author Contributions:** Writing—review and editing, X.L.; resources, Y.L.; software, S.T.; supervision, L.W. All authors have read and agreed to the published version of the manuscript.

**Funding:** This research was funded by the National Natural Science Foundation of China (51179116) and the Natural Science Foundation of Shanxi province (202303021211141).

**Data Availability Statement:** Not applicable.

**Conflicts of Interest:** The authors declare no conflict of interest.

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
