# Peer review of "A Study on the Measurement Characteristics of the Spring-Plate Flow Measurement Device"

_water, doi:10.3390/w15112092_

Round 1

Reviewer 2 Report

The scientific article "A study on the measurement characteristics of the spring-plate flow measurement device" is devoted to the actual problem of the use of open channel flow measurement devices that are accurate and easy to carry out. 

The paper title clearly describes the content of the work. The paper is well-written and the quality of the figures is acceptable. The paper is interesting with some valuable conclusions and is recommended for possible publication in Sustainability journal. The article contains extensive experimental material. The experiments are carried out with a high degree of reliability and are of great scientific value. However, some revisions should be performed. 

1. What are the limitations of this study? What is the limitation of using a spring-plate flow measurement device?

2. Mesh independency should be proved because it can be seen when the authors have parts of the different mesh they have different velocity distributions. What were the criteria for the final grid density?

3. How the inlet turbulence values have been chosen? 

4. What about y+? Better elaborate on its influence.

5. The measured minor loss is usually given as a ratio of the head loss hm=dp/(ro*g) through the device to the velocity head V2/(2g) of the associated piping system. Loss coefficient K=hm/(V2/(2g)=dp/(0.5*ro*V2). So, I think, that authors should consider classic determining of minor(local) losses instead of "Head loss ratio"

In general, I think that a few improvements could be made to elevate the quality of the manuscript.

Reviewer 3 Report

Dear Editor,

The reviewer reviewed the manuscript (MS) titled “A study on the measurement characteristics of the spring-plate flow measurement device” submitted to Sustainability journal by Li et al. for potential publication in detail to meet the scientific requirements.

In the manuscript, the authors conducted a study by fitting flow rate and force and analyzed some hydraulic characteristics by theoretical analysis and numerical simulations.

General comments are listed below:

Manuscript should be improved regarding grammatical/writing mistakes. For instance, “as” (“not with increasing”, “as….increases” may be preferred) should be preferred when using increasing or decreasing trends.

Use of punctuations should be checked.

Some sentences are too long to follow easily.

Reference numbers should be given after references (for instance, et al.).

Materials and methods and numerical findings should be mentioned in Abstract.

What is contribution of this study? Motivation is understandable, but novelty of the manuscript is not clear. It should be clearly revealed and outstanding side of the study should be clearly presented by discussing findings of similar researches.

A flowchart would be better for efficient presentation.

Are the results independence of mesh sizes or sensitive to mesh sizes? This is an important problem to achieve results independent of mesh sizes. What about outputs if the mesh sizes change?

Initial and boundary conditions should be presented as a table.

Discussion and conclusion section should be separated and avoided summarization.

Manuscript should be improved regarding grammatical/writing mistakes. For instance, “as” (“not with increasing”, “as….increases” may be preferred) should be preferred when using increasing or decreasing trends.

Use of punctuations should be checked.

Some sentences are too long to follow easily.

Round 2

Reviewer 3 Report

What is contribution of this study? Motivation is understandable, but novelty of the manuscript is not clear. It should be clearly revealed and outstanding side of the study should be clearly presented by discussing findings of similar researches. The authors presented these issues in Discussion section. Novelty and contributions are mentioned in Introduction section.

Discussion section must be located before Conclusion section. Comprehensive discussion is required in the manuscript.

Round 3

Reviewer 3 Report

It can now be accepted for publication.

Author Response

We are very grateful for your approval of this manuscript. Thank you very much for your comments and suggestions.